# Multicenter Observational/Exploratory Study Addressed to the Evaluation of the Effectiveness and Safety of Pharmacological Therapy in Opioid-Dependent Patients in Maintenance Therapy in Southern Italy

**DOI:** 10.3390/pharmaceutics14020461

**Published:** 2022-02-21

**Authors:** Fatima Maqoud, Giada Fabio, Nunzio Ciliero, Marina Antonacci, Francesca Mastrangelo, Giorgio Sammarruco, Roberto Cataldini, Gabriella Schirosi, Salvatore De Fazio, Domenico Tricarico

**Affiliations:** 1Department of Pharmacy-Pharmaceutical Sciences, Pharmacology, University of Bari, Via Orabona 4, 70125 Bari, Italy; fatima.maqoud@uniba.it (F.M.); nunzio.ciliero@studenti.uniba.it (N.C.); marina.antonacci@studenti.uniba.it (M.A.); 2Ser.D. Bari, ASL-BA, Via Amendola, 124/C, 70126 Bari, Italy; g.fabio@asl.bari.it; 3Ser.D. Lanciano, ASL2 Lanciano-Vasto-Chieti, Via Martiri Lancianesi 17/19, 66100 Chieti, Italy; francesca.mastrangelo@asl2abruzzo.it; 4Ser.D. Poggiardo, Piazza Partigiani, s.n.c., Poggiardo, 73037 Lecce, Italy; g.sammarruco@asl.lecce.it; 5U.O. Double Diagnosis Gallipoli, Via Marconi, 1, 73014 Gallipoli, Italy; r.cataldini@asl.gallipoli.it; 6Ser.D. Taranto, Contrada “Rondinella” c/o Ospedale “Testa” Strada Statale 106, 74100 Taranto, Italy; g.schirosi@asl.taranto.it; 7Ser.D. Mesagne Piazza Gioberti 1, 72023 Brindisi, Italy; s.defazio@asl.brindisi.it

**Keywords:** levomethadone, abuse substance, clinical study, QTc, double diagnosis

## Abstract

A multicenter-observational study was performed to assess the effectiveness of rac-methadone, levomethadone, and buprenorphine in opioid-dependent patients in polytherapy in Southern Italy. The primary endpoint was the reduction of urinary positivity to the substances and the maintaining doses. Patients (N = 266, age = 44.80 ± 5.65, male = 79.70%, female = 20.30%) have been recruited. At recruitment, 75% of them were on treatment with rac-methadone, levomethadone, and buprenorphine/naloxone. The patients were grouped into three clusters. The levomethadone patients of Cluster A (N patients = 211), after 180 days, showed stability in urinary methadone positivity, with a marked decrease in heroin −53 ± 4%, cannabinol’s −48 ± 2%, and cocaine −37 ± 6% positivity, with no differences between treatments. A lower QTcF value of 426 ± 8.4 ms was recorded in the levomethadone patients (delta = −19 ms) vs. rac-methadone, at significantly lower doses of levomethadone (−34%, −50.2% in males) (*p* < 0.05). The Cluster B data were collected from 37 patients, with a high prevalence of comorbidity infections (HIV/HCV/HPV), monitored for 21 months during COVID-19. High doses of levomethadone (58.33 ± 31.58 mg/day) were needed to stabilize those that were negative for opioids and cannabinoids, in contrast to the rac-methadone and buprenorphine/naloxone patients that showed positive toxicology. Eighteen patients of the Cluster C in double diagnosis (major depressive 38.90%, bipolar 27.78%, and schizophrenia 16.67%) were stabilized with high doses of racemate 97.5 ± 8 mg/day, 51.8 ± 5 mg/day of levomethadone (−46.8% vs. rac-methadone; −71% in men), and 2.5 ± 1 mg/day of buprenorphine/naloxone. Three patients in remission were treated with tapering doses of levomethadone. Significantly reduced QTcF values were recorded with levomethadone (delta −32 ms vs. rac-methadone) in the bipolar patients, as well as the schizophrenia patients in remission (delta −45.19 ms vs. rac-methadone). Our patients were safely stabilized. Levomethadone, compared to the racemate, contributes to reducing the illicit use, especially of opioids and cannabinoids at significantly lower doses with cardiovascular safety, which, in bipolar patients, is clinically significant.

## 1. Introduction

With the term “drug addiction”, the WHO (World Health Organization) defines a “chronic and relapsing disease that pushes the individual, in a forced manner, to take substances at increasing or constant doses to have temporary subjective beneficial effects, whose persistence is inextricably linked to the continuous intake of the substance”. In the DSM-5 (Diagnostic and Statistical Manual of Mental Disorders), the category “addictive and substance use-related disorders” has experienced substantial changes from previous editions of the DSM: The categories of “substance abuse” and “substance dependence” have been reunified into a single disorder, measured on a continuum mild to severe, whose criteria for diagnosis (almost identical to the previous criteria) were merged into a single list of 11 symptoms. The DSM-5 sets the following conditions for the diagnosis of a substance use disorder (DUS) [1]:*Tolerance*: A phenomenon for which it is necessary to intensify the behavior of use (for example, by increasing the amount of drug to be used or frequency of intakes), in order to achieve the same effects on the body.*Withdrawal*: characterized by the presence of emotional or physical symptoms that occur when the subject cannot implement the behavior of intake.*Interruption or reduction of social, work, or recreational activities*: The use of drugs and onset of the disorder cause a series of damage to the functioning of the person who uses it (conflicts with affectively important people, work problems, influences on self-esteem, etc.) that increase in intensity, progressively harming the patient.*Unsuccessful attempts to reduce and control use*: it is frequent that the patient, before formally seeking help from the psychologist or services, has tried on his own to reduce the use or “control” it. Generally, a phase is observed, in which the patient is firmly convinced that he can limit his conduct on his own by creating a mode of use that can be *reconciled* (but only ideally) with the rest of his life, commitments, and duties.*Expenditure of time*: when the disorder is established, or being established, a criterion to look at is that of the time that the patient devotes to research, use, or recovery from the effects of the substance. The more the addiction is over, the greater the time that is dedicated to the substance in a day, until it becomes the only activity present in the most serious cases.*Loss of control over use*: The pathological behavior of use of the substance tends to occur, despite the negative consequences that it has brought over time and person’s awareness of it (the behavior of use becomes “compulsive”).*Continuous use despite the awareness that the drug is a problem*: many patients do not stop, even in the face of the onset of serious health risks or clear family crises.*Recurrent use with the inability to fulfill their duties*: many patients lose their jobs due to drug intake, interrupt the course of study, or become unable to perform their family or parental duties.*Use in situations at risk*: over time, the ability to estimate the risk associated with hiring is progressively reduced; becoming compulsive assumptions, it can happen to feel “forced” to make abuse, despite having to drive or perform precision tasks that cannot be “rationally” reconciled with the state of alteration given by the substances of abuse.*Recurrent use despite this causes social or interpersonal problems*: As previously stated, drug use becomes salient, even to the detriment of one’s effective relationships.*Craving*: urgent desire for the substance.

Recently, it has been reported that about 62 million people used opioids for non-medical purposes in 2019, corresponding to 1.2% of the world’s population. North America (3.6%) has the highest prevalence rate, and Europe the lowest (0.8%). The illegality of opioids, such as heroin, precludes the accurate assessment of how many people use these drugs [2]. Illicit opioid addiction is a significant public health problem, especially when you consider that heroin use is also associated with the spread of infectious diseases (e.g., HIV, hepatitis B, and C) and overdose deaths [3]. Drug use in Europe compromises of a wide range of substances. Among the people who use drugs, poly abuse is common, but difficult to estimate, and individual patterns of use range from experimental to habitual to addiction. The prevalence of high-risk opioid use among adults (15–64 years) in 2019 is estimated to represent 0.35% of the EU population, equivalent to 1 million high-risk opioid users. In 2019, in the European Union, there were 510,000 users in substitution therapy for opioid addiction. Opioid users account for 26% of drug treatment claims. Opioids resulted in 76% of the overdose deaths reported in the European Union in 2019. National prevalence estimates range between less than one and more than seven high-risk opioid users per 1000 inhabitants aged 15 to 64. Overall, this translates to around 0.35% of the European population, or 1 million high-risk opioid users in 2019. The four most populous countries in the European Union (Germany, Spain, France, and Italy) account for more than two-thirds (68%) of this estimate. In 2019, opioid use was reported as the main reason for access to specialist drug treatments by 84,000 users, accounting for 27% of all those who underwent treatment in Europe. Of these, about 20,000 were accessing treatment for the first time. Heroin was the primary drug for 13,600 (79%) of the 17,300 new users taken in charge who had reported the opiate, a relatively stable number, compared to the previous year [4]. In Italy, during 2020, the Ser.D. (Servizi Dipendenze) assisted a total of 125,428 drug addicts, for which 12% were new users. A total of 86% of Ser.D. patients were male, with an average age of 41 years. Female users, on the other hand, have an average age of 40 years. New users are younger (on average 9 years) than those already in charge. On average, the subjects present in treatment at the socio-rehabilitative structures, surveyed in 2020, were 13,781 per day. A total of 35% of users were in treatment for the primary use of cocaine/crack, with an equal share for heroin/opiates. At the end of 2020, there were 14,148 drug-addicted prisoners in prison, equal to 26% of the entire prison population—a decrease, compared to the previous two years. The main activity provided by the services is individual *counseling* on the risks of infectious diseases, followed by *hepatitis screening* campaigns and the distribution of information material [5].

All this is in collaboration and synergy with therapeutic communities, municipal administrations, and volunteering. Alongside these services, there is the work carried out at the Dual Diagnosis Operating Unit.

The term double diagnosis has been adopted to indicate a particular clinical condition, in which the picture of drug addiction or abuse (e.g., heroin, cocaine, alcohol, or drugs) and a psychiatric disorder coexist [6].


*Double diagnosis is defined by the WHO as the “coexistence in the same individual of a disorder due to the consumption of psychoactive substances and another psychiatric disorder”*


The condition of patients with *double diagnosis* is particularly serious and requires networking between local psychiatric services, hospital services, and possibly private specialists. The patient shows, compared to others:a higher relapse rate;more frequent hospitalizations;more likely to commit crimes and end up in prison;higher risk of contracting infectious diseases, such as HIV and hepatitis.

In this group, a reference is made to types of patients who are also very different from each other. Some develop a psychiatric disorder, secondary to substance abuse. They may be patients who have developed anxiety and mood disorders, secondary to the condition of abuse or consequences of abuse (loss of work, social and family relationships, etc.). In addition, as well-documented by neuroimaging studies, the use of drugs can create irreversible damage to the central nervous system, with consequent psychic and behavioral problems, often very serious. The second group of patients may have a primary psychiatric diagnosis before substance abuse. The state of psychic suffering can, in these patients, represent a risk factor for developing a pathological addiction. The use of substances can also be interpreted as an attempt, by these patients, to self-care for the primary psychiatric disorder [7]. A third group, on the other hand, has a psychiatric disorder and substance abuse or addiction in parallel. In these cases, it is *“as if”* the patient develops the two disorders independently. Dual diagnosis treatment is necessarily an integrated multi-step treatment that provides both detoxification/withdrawal and prevention of the risk of withdrawal and craving, and treatment of the psychiatric disorder in comorbidities.

Drug replacement therapy is considered the most effective treatment for opiate-dependent patients. However, issues related to substance abuse addiction, such as psychiatric comorbidities, infectious diseases, psychological distress, and stigmatization, can make treatment largely challenging and often requires a multidimensional approach to therapy [8].

Morphine receptor agonists and partial agonists represent the most recognized drug treatment in opiate-dependent patients worldwide [9]. Numerous clinical studies and evaluation programs demonstrate its effectiveness in reducing the illicit use, related mortality, and transmission of blood viruses, human immunodeficiency viruses (HIV), hepatitis C (HCV), and hepatitis B (HBV). Therapy also improves physique, psychological health, and quality of life and promotes social functioning and reintegration [10,11]. However, side effects can affect the patient’s response and reduce the maintenance rate [12].

Rac-methadone is the reference drug in the treatment of opioid dependence. It is considered the most effective of the drugs available in flexible-dose, in combination with psycho-social interventions, resulting in better retention in treatment. The prescription of these drugs is based on physical examination and clinical, drug, and psychological history. Rac-methadone is proposed in patients with a double diagnosis of addiction and maniacal psychosis, even borderline, who need sedation. However, the interindividual response to rac-methadone is strongly variable, ranging from a minimum initial dose per os of 15–20 mg/day for the treatment of abstinence symptoms to a dose of 40 mg/day in methadone-dependent patients, at average doses of 50–120 mg/day, up to prescribed doses, in off-label, of 300 mg/day, in the maintenance of severely dependent patients in some areas of Puglia. The response is influenced by various factors. One of them is related to molecular structure. Methadone is, in fact, a racemic mixture, in which the levo enantiomer is responsible for most of the therapeutic effects, while there are side effects [13]. Other factors that come into play are environmental ones and those related to the concomitant use of substances of abuse and drugs, which interfere with the metabolism and pharmacodynamics of the methadone itself. Highly effective doses of raceme methadone and long-term therapy, in patients in maintenance, expose them to adverse reactions, such as QT tract lengthening, respiratory depression, liver disease, infections, fatigue, constipation, hypertension, and hypotension, increased moderate-to-severe intracranial pressure, and sudden death.

Buprenorphine, on the other hand, has a well-known, different pharmacological profile, compared to methadone. It is a partial agonist of the opioid μ receptor: it mimics the pharmacological effect of an opioid, but to a much lesser extent, thus preventing opioid withdrawal symptoms. However, it has a slow rate of dissociation from the receptor, thus having a prolonged duration of action, compared to other opioids. Instead, it has antagonistic activity on k receptors, which contributes to the dysphoric and psychotomimetic effects of opioids. Buprenorphine reduces these effects. The partial competition of opiate receptors makes the molecule a valid alternative to methadone, as the μ risk of respiratory depression from overdose is greatly reduced [14,15,16,17,18]. This is especially important in the drug addict population. It is also administered sublingually, in combination with the antagonist naloxone, in a single administration/day, at doses even greater than 2 mg/day. The average daily maintenance dose is 8 mg. The majority of patients do not require doses greater than 16 mg/day. The efficacy and safety of buprenorphine tablets have been demonstrated in clinical trials at doses of up to 24 mg/day. Recent reports propose buprenorphine as the drug of choice in subgroups of dual-diagnosed patients with severe opioid dependence, tolerance, and major depression [19].

The clinical activity of buprenorphine/naloxone is mainly based on the different pharmacodynamic properties of the two drugs contained therein. In particular, when taken correctly, and this is sublingually, only the absorption of buprenorphine occurs with consequent clinical efficacy (naloxone is poorly absorbable enterally), while, if taken incorrectly intravenously, in addition to buprenorphine, naloxone is also absorbed, which acts as a receptor antagonist, generating a clinical situation of discomfort and/or mild withdrawal in the patient, such as to discourage its use by this route. Naloxone has long been known as a short-acting opioid receptor antagonist. It binds to receptors with high affinity in a competitive manner. It is administered in low doses and can reverse the side effects of opioids, such as respiratory depression, sedation, and hypotension, without significantly affecting analgesia. At high doses, however, naloxone can block opioid analgesia, causing withdrawal. However, it presents clinical activities only for parenteral administration.

The levomethadone-based drug is 10 times more active, in vitro and in vivo, on opioid receptors than methadone raceme in pre-registration studies. This is the latest drug in this therapeutic to be available in Italy in 2015. The effects, related to the elimination of opioid symptoms and compulsive desire to search for the substance, as well as the analgesic effect, are attributed to the levomethadone enantiomer [20]. In contrast, dextro-methadone is responsible for adverse effects, including cardiotoxic effects. This drug shows stereoselective pharmacodynamics and pharmacokinetics that justify its use, instead of rac-methadone, in patients at high cardiological risk. The effective maintenance dose of levomethadone hydrochloride may be up to 60 mg/day. In patients already receiving methadone, the dose indicated for substitution from the racemate to the R enantiomer is expected to be −50% [21].

Opioid receptors can form hetero-multimeric complexes, with receptors for cannabinoids CB1, or glutamate N-Methyl-D-Aspartate (NMDA) receptors with important functional implications of cross-talk between systems. Methadone inhibits NMDA receptors in an enantio-selective manner, with relevant implications in neuroprotection, pain control, and depression. NMDA receptors, such as NR1/2A and NR1/2B, show a higher affinity for levomethadone than morphine and dextro-methadone [22,23].

Rac-methadone undergoes a significant first-pass effect after absorption with hepatic metabolism, which transforms it by N-demethylation by the cytochrome P450 enzyme system (CitP) and, in particular, by *CitP3A4* into EDDP (2-ethylidene-1,5-dimethyl-3,3-diphenylpyrrolidine), the inactive metabolite, and EMDP (2-ethyl-5-methyl-3, 3-diphenylpyrroline). In addition, *CitP2D6* and *CitP1A2* (inducible by Smoke) [24] are involved in the metabolism of rac-methadone raceme. The enzyme system of *CitP2C19* metabolizes levomethadone, while dextro-methadone is metabolized by CitP2B6. Serious adverse reactions from rac-methadone overdose, such as respiratory depression and ventricular arrhythmia, are associated with gene polymorphisms [25]. *CitP2D6* gene variants 3, 4, and 5 are associated with the slow metabolization of rac-methadone, resulting in accumulation. The polymorphism of the *CitP2B6* *6/*6 gene is associated with respiratory depression and death from rac-methadone overdose. On the *CitP2B6 *6/*6* slow metabolizers, the risk from dextro-methadone is significantly increased, compared to levomethadone. Gene polymorphisms of *KCNH2*, encoding the hERG potassium channel, are associated with QTc tract prolongation [25,26,27].

To date, limited data on the effectiveness of levomethadone are available in clinical practice at the national level and Apulian reality. There are no national and regional guidelines or shared indications that help the clinician in prescribing these drugs to patients in maintenance therapy. The success of drug therapy depends on psycho-social support, which is, in turn, regionalized so foreign experiences cannot be easily transferred to the site. The phenomenon of immigration has then exacerbated the medical–social problem. The prescription of these drugs and dosage is empirically based on the experience of the clinician, as well as the patient’s condition: mild, moderate, or severe tolerance to opioids, as well as from psychiatric and psychological supportive care. Although the supporting data indicate that the best treatment of drug addiction requires an integrated approach, in case of double diagnosis, these patients are followed in psychiatry and at drug addiction centers, as well as in the case of adverse reactions in internal medicine departments with loss of dose control and low quality of intervention. The resulting data will be compared with the international reality.

## 2. Materials and Methods

### 2.1. Ethical Consideration

The data were pooled from different centers participating to the study. The study protocol n° 5869, entitled “Studio Osservazionale/Esplorativo Prospettico Multicentrico Indirizzato Alla Valutazione Dell’efficacia E Sicurezza Della Terapia Farmacologica In Pazienti Dipendenti Da Oppioidi In Mantenimento: Fattori Che Influenzano La risposta alla Terapia Nell’aerea Sud Est Meridionale”, was approved on 28 November 2018 by the Ethical Committee of Az. Policlinico Ospedaliero Universitaria Consorziale dell’Università degli Studi di Bari, Italia, under the scientific responsability of the Prof Domenico Tricarico (P.I. of the project). Data collection, analysis, reports, and data privacy were in agreement with Good Clinical Practice and Helsinki Declaration. All patients were informed about risk and signed the Informed Consent.

### 2.2. Protocol

The primary endpoint of effectiveness was the temporal reduction of urinary positivity to the substance of abuse, expressed as a percentage of responders out of the total number of recruits and dosing regimens. Four drug treatments were compared: levomethadone, rac-methadone, buprenorphine, and buprenorphine/naloxone. Secondary endpoints were the safety assessment during poly therapies for different patient cohorts, collected on the case report form (C.R.F.), following the Medical Dictionary for Regulatory Activities (MEDRA). An electrocardiogram (ECG) was performed, and the heartbeat (HB), the QT intervals of the ECG trace, were measured and corrected by the Fridericia formula (QTcF). Psychopathology was assessed by the Addiction Severity Index (ASI) and Symptom Checklist-90 (Scl90) and other functional tests, when available in the specific centers.

The patients were divided in three clusters, on the basis of the temporal collection of the data and diagnosis. The Cluster A patients (N patients = 211) were enrolled at the following centers, from 2018 to 2019, for six months: Ser.D. Bari Puglia, Dott. Antonio Taranto, Dott.ssa Giada Fabio, Ser.D. Chieti Abruzzo, Dott.ssa Paola Fasciani, Ser.D. Lanciano (CH) Abruzzo, Dott.ssa Francesca Mastrangelo, Ser.D. Poggiardo (LE) Puglia, Dott. Giorgio Sammarruco, Ser.D. Taranto Puglia, Dott.ssa Vincenza Ariano, Dott.ssa Gabriella Schirosi, Dott.ssa Maria Rita Quaranta, and Ser.D. Manfredonia (FG) Puglia, Dott. Angelo De Giorgi.

The Cluster B patients (N patients = 37) were enrolled at Ser.D. Mesagne (BR) Puglia, Dott. Salvatore De Fazio, and Dott.ssa Rita Campana from 2020–2021, during the COVID-19 period, for 21 months.

The Cluster C patients (number of patients = 18) were enrolled from 2018 to 2021 at the U.O. Double Diagnosis of Gallipoli (LE) Puglia, Dott. Roberto Cataldini.

Inclusion criteria were: Adult patients > 18 years of various ethnic origin, under opioid-dependence; patients in maintenance with methadone raceme, levomethadone, buprenorphine, or buprenorphine/naloxone; patients on drug treatment for comorbidities in the dual diagnosis of drug dependence and psychiatric disease; HIV/HCV/HBV-infected patients, as well as hypertensive, cardiopathic, and diabetic patients; treatment-naïve patients; patients dependent on other substances of abuse.

Exclusion criteria: patients unable to cooperate and provide informed consent. Patients who, at the time of enrollment, are in any pathological condition, of severe degree in disease progression, that requires hospitalization.

### 2.3. Statistical Analysis

The analytical data will be expressed as mean ± S.D., and median, minimum, and maximum dose values were taken. The calculation of the sample size was carried out on the primary outcome using ANOVA one-way (G*POWER software 3.1) using an input power value of 0.85. Statistical difference between means was evaluated by Student *t*-test for *p* < 0.05.

## 3. Results

The results of this work were based on 266 patients, 75% were on treatment (most of them for less than 1 year) with rac-methadone (56.11 ± 9.2 mg/day), levomethadone (36.93 ± 5.7 mg/day), and buprenorphine/naloxone (6 ± 0.9 mg/day), and the patients were divided into three clusters (Table 1, Figure 1A).

The Cluster A patients (N patients = 211) enrolled at different centers at recruitment, and the 75% of patients were on maintenance therapy; most of them, about 80%, had been on for less than 1 year. Most of the patients were under rac-methadone treatment at the first visit (Figure 1B,C) and were evaluated after 180 days. The comorbidity study in males and females showed that some patients had psychiatric, infectious, cardiological, and metabolic comorbidity (Table 2 and Table 3).

The daily dosages remain constant after 180 days for all treatments, as well as VAS. There is, instead, a gender difference in the average dose of levomethadone; women were, indeed, stabilized at a dosage greater than +56.90%, compared to the opposite sex (Δ = +16.36 mg/day), and males were stabilized at a significantly lower dose of levomethadone (Δ = −28.46 mg/day, −50.2%) vs. rac-methadone (*p* < 0.05), while females was stabilized at higher doses of levomethadone (Δ = −9.8 mg/day, −18%) vs. rac-methadone.

The drug treatments were effective in reducing the toxicological parameters; indeed, the urinary positivity to abuse substances was reduced after the 180 days of treatment, with no differences between groups, despite an unbalance in favor of levomethadone vs. other treatments. The treatments appear to be less effective in reducing cocaine positivity (Table 4).

We also evaluated the QtcF of 123 patients at recruitment and after 180 days of treatment, and we found an unbalance, in terms of the reduction of this safety parameter, in favor of the levomethadone group vs. other treatments. The observed changes were not statistically significant, but were clinically significant, at recruitment (Δ= −19 ms levomethadone vs. rac-methadone) and after 180 days (Δ= −13.1 ms levomethadone vs. rac-methadone). No adverse cardiovascular effects were observed (Table 5).

The following adverse drug reactions (ADRs) were observed: mild/moderate grade of constipation and muscle fatigue in all treated groups, which were considered expected reactions, with no safety concern and differences between groups after the 180 days of treatment; one case of Stevens–Johnson syndrome was, however, observed with Rac-methadone that was transmitted to the Italian pharmacovigilance database (VigiFarmaco). Adverse events of any nature were: decreased willpower, job loss, increased alcohol use, and changes in personality and mood, including dullness and depression, which were similarly distributed among patient groups.

Therefore, the levomethadone maintenance treatment was equally effective in improving the urine toxicology and QTcF in the treated patients, showing, in addition, a clinically relevant reduction of this ECG parameter at the first visit (Δ = −19 ms levomethadone vs. Rac-methadone) and after 180 days (Δ = −13.1 ms levomethadone vs. rac-methadone), but at much lower doses than rac-methadone—in particular, in male patients.

The Cluster B data were collected from 37 patients, for 21 months at 90 days intervals, starting from 01 January 2020, during the COVID-19 pandemic. The focus was on the variation of maintenance therapy, dosages, and urinary positivity to substances of abuse (opioids, cannabinoids, and cocaine) during the time. These patients were high-risk for adverse events, being under criminal prosecution (Table 6), and affected by infective diseases (35%) (HIV/HCV/HPV). These patients were stabilized at high doses of drugs, including levomethadone. The maintaining doses at recruitment were rac-methadone (65 mg/day), levomethadone (60.00 mg/day), and buprenorphine/naloxone (6.67 mg/day), and the preferred option was rac-methadone. The patients had all already been on treatment, on average, for over 5 years.

After 1 year and 9 months of observation, we had seven changes in the cohort (including six therapy changes):One patient (woman) switched positively from rac-methadone to levomethadone, until remission;One patient (woman) receiving rac-methadone discontinued therapy;Four patients (men) switched from rac-methadone to levomethadone;One patient (men) switched from levomethadone to rac-methadone

The number of patients receiving rac-methadone (−25%) decreased, compared to an increase with levomethadone (+50%), which became the preferred therapy. The mean dose values were not affected during treatments (Table 7).

There is obvious negativity to opioids, given by levomethadone, even after 21 months of treatment (Figure 2A). Rac-methadone gave good results, bringing a decrease of −50% of positivity. Levomethadone, together with buprenorphine/naloxone, also confirms the negativity towards cannabinoids (Figure 2B). Rac-methadone confirms the benefit in decreasing positivity, also towards cannabinoids (−33%), as well as for the positivity to cocaine, referable to all treatments, without distinction. This could suggest a minimal, if not absent, effect on the influence in cocaine use, as opposed to cannabinoids and heroin, where levomethadone is significantly more effective than rac-methadone in reducing or discouraging their use (Figure 2B).

To evaluate if the observed effects were stably maintained over all of the time during the 21 months, we monitored these data at 90 days intervals. For further clarity of analysis, patients were separated into two subgroups: those who did not change medication (in maintenance) and those who did switch to a new therapy. The comparison between the three treatments in patients (number of patients = 30) shows an almost constant dose of buprenorphine/naloxone. Rac-methadone, after an initial increase in dose, returned to the initial mean value. The dose of levomethadone remained constant for more than a year and then decreased (Figure 3).

It is highlighted that the decrease in urinary positivity to opioids in patients in rac-methadone maintenance is constant over time (Figure 4A). Rac-methadone decreases opioid positivity over time, while not eliminating it. In patients with buprenorphine/naloxone, there is almost a quarterly alternation between positivity and negativity to opioids. Levomethadone, on the other hand, confirms its effectiveness in maintaining constant opioid negativity over time (Figure 4A). Cannabinoid positivity is almost constant over time with rac-methadone (Figure 4B). Both levomethadone and buprenorphine/naloxone are effective in maintaining negativity towards cannabinoids, as well. High variability of cocaine positivity is observed over time. This suggests that there are no specific effects of these drugs on cocaine addiction (Figure 4B), as already observed in the patients of Cluster A.

Seven out of thirty-seven patients, of Cluster B, changed their treatments over the observation period. Analyzing the individual data, there emerges, following the previous graph, an increase in levomethadone doses in patients who switch from the raceme (Figure 5A). For others, however, after an initial “adjustment” of the dosage, this remains constant. Additionally, for raceme methadone, this increase is noted immediately after the switch.

The switch did not seem to vary opioid positivity, except in the last period of observation (Figure 5B), while there was an annulment of cannabinoid positivity after 1 year from the switch to levomethadone (Figure 5C). Even those who return to the racemate do not show any new positivity. The switch seems to not affect changing urinary positivity to cocaine, as observed in the Cluster A patients (Figure 5D).

The Cluster C patients (number of patients = 18) were treated at l’U.O. Double Diagnosis of Gallipoli (LE) (Dr. Roberto Cataldini). These were mostly males (Table 8) who were diagnosed as drug addicts and mostly affected by major depression and bipolar disorders, treated with levomethadone and buprenorphine/naloxone, some of them until detox (Figure 6A,B). The male patients were stabilized at lower doses of levomethadone vs. rac-methadone (−71%) (Table 8). They received a complete diagnosis of psychiatric disorder, and were treated with a balance between drugs that act on the central nervous system and for the therapy of detox from substances of abuse. Particular attention has been paid to this cluster, since there are as many as three patients in remission after treatment with levomethadone. In addition, two patients had previously “switched” from rac-methadone to levomethadone and were still being treated. Data on SCL90, such as GSI and PSDI, agree between therapies, except for buprenorphine, which scores higher than average. Patients in remission, after levomethadone therapy, had a lower PST score than those who are still on treatment (Table 9). These patients received concomitant antipsychotic and/or antidepressant drugs (Table 10). In Cluster C, there was greater use of levomethadone, which was the preferred therapy.

Levomethadone, however, does not appear to have a marked effect in avoiding substance abuse positivity in this patient cluster at this low dose. The result is highlighted, instead, with high doses of rac-methadone, since the patients treated with this drug do not show positivity to drugs (Figure 6C).

Clinically and statistically significant reduced QTcF values were observed with levomethadone (Δ −32 ms vs. rac-methadone) in bipolar patients (Figure 7A), as well as in schizophrenia patients in remission (Δ −45.19 ms vs. rac-methadone) (ANOVA one-way). No cardiovascular adverse effects were observed with these drugs (Figure 7B,C).

## 4. Discussion

In the present work, we showed that, in our Ser.D., the recruited Italian Caucasian patients (N patients = 266), made up of 79.70% males and 20.30% females, were safely stabilized and maintained under treatment with rac-methadone (56.11 ± 9.2 mg/day), levomethadone (36.93 ± 5.7 mg/day), and buprenorphine/naloxone (6 ± 0.9 mg/day). There is, indeed, strong evidence that longer treatment episodes, with high retention and maintenance of this opioid use disorder, are associated with better clinical and social outcomes, despite treatment completion [27,28]. The patients were grouped into three clusters, based on the period of recruitment and comorbidity conditions.

The patients of Cluster A (N patients = 211), after 180 days of observation, were stabilized at significantly lower doses of levomethadone (−50.2%, 28.75 ± 7.5 mg/day in males) vs. rac-methadone, and the levomethadone patients showed stability in urinary methadone positivity, with a marked decrease in heroin −53 ± 4%, cannabinol’s −48 ± 2%, and cocaine −37 ± 6% positivity, with no differences between groups. A clinically lower QTcF value of 426 ± 8.4 ms was also recorded in the levomethadone patients (Δ = −19 ms) vs. rac-methadone, however, at significantly lower doses of levomethadone (−34%) vs. racemate.

The Cluster B data were collected from 37 patients, monitored for 21 months, during COVID-19. These patients were high-risk for adverse events and affected by infective diseases (HIV/HCV and HPV) (45% prevalence). High doses of levomethadone (58.33 ± 31.58 mg/day) were needed to stabilize these patients, which showed negative urine toxicology to opioids and cannabinoids, in contrast to rac-methadone- and buprenorphine/naloxone-treated patients, which showed positive urine toxicology. Levomethadone became the preferred therapy in these patients. Maintenance treatments did not affect reducing urinary positivity to cocaine. It should be of note that the dosing of levomethadone and rac-methadone are affected by drug interactions with antivirals, which are known to induce the CYP3A4-mediated metabolism of these drugs, explaining the high dosing needed in the Cluster B patients, in which the prevalence of the infective (HIV/HCV) diseases was elevated vs. Clusters A or B. Additionally, the combination of cocaine and cannabinoids can reduce the serum levels of these drugs [29].

Cluster C consists of 18 patients with a double diagnosis of major depressive disorder (38.90%), bipolar disorder (27.78%), and schizophrenia (16.67%). These were stabilized with high doses of racemate (97.5 ± 8 mg/day), 51.8 ± 5 mg/day (−46.8% vs. rac-methadone; −71% in man) of levomethadone, and 2.5 ± 1 mg/day of buprenorphine/naloxone. We had three patients in remission that were treated with tapering doses of levomethadone. Significantly reduced QTcF values were recorded with levomethadone (delta −32 ms vs. rac-methadone) in bipolar and schizophrenia patients in remission (delta −45.19 ms vs. rac-methadone). Our patients were stabilized and safely treated in our service. Levomethadone, compared to the racemic, contributes well to reducing the illicit use, especially of opioids and cannabinoids, at significantly lower doses, with cardiovascular safety, which, in some bipolar patients, is clinically significant.

Levomethadone, was superior, compared to raceme and buprenorphine/naloxone, in reducing the urine toxicology of opioids and cannabinoids, as well as counteracting the illicit use of opioids, in particular, but also of cannabinoids, even during the pandemic period, however, at high doses that were comparable with that of rac-methadone. Levomethadone, however, does not appear to have a marked effect in avoiding substance abuse positivity in dual diagnosis cluster at low doses. The result is highlighted, instead, with high doses of rac-methadone, since the patients treated with this drug do not show positivity to drugs. The maintenance therapy was not effective against cocaine addiction, in terms of urine toxicology.

On the other hand, anti-depressive and antipsychotic drugs are known inhibitors of the CYP3A4- and 2D6-mediated metabolism of these drugs, explaining the low dosing needed in the Cluster C patients with psychiatric comorbidity.

Patients in remission (treated with levomethadone) had the lowest score in PST (−29.75, compared to the average). With buprenorphine, there were higher values in PST (+15.25) than the average. With buprenorphine/naloxone, instead, there was a decrease in PST (−27.25), compared to the average.

Lower values in QTc confirm the safety, from a cardiological point of view, of levomethadone vs. rac-methadone, which, in some bipolar patients and those with major depressive disorder, is clinically significant. They have a lower QtcF of 32 and 13.5 ms, respectively, than those taking methadone at significantly lower maintenance doses.

The take-home dosage procedure, applied during the pandemic with contrasting expert opinions/experiences [30,31], did not affect patient stability in our setting, despite the increase of the mean doses of all drugs, especially for rac-methadone, that we observed, in line with some reports [32]. The coronavirus pandemic (COVID-19) has introduced many concerns surrounding access to opioid treatment. In March 2020, the Substance Abuse and Mental Health Services Administration (SAMHSA), U.S.A., issued guidance allowing for the expansion of take-home methadone doses. This led to a near-doubling of take-home rac-methadone doses during the COVID-19 exemption period; the increase in take-home doses was not associated with negative treatment outcomes in methadone-adherent clients [32]. No changes in the rate of overdose cases, in our setting, were reported, in line with the recent report [33].

The effectiveness of levomethadone, in maintenance therapy, in our setting, is confirmed by the fact that, to date, all patients in remission in our centers had been on therapy with this drug. The transition from higher doses of rac-methadone is possible, as was applied in our centers. Several dosing strategies have been proposed to soften withdrawal symptoms and facilitate transfer, including the use of other opioids or medications and, especially, micro-dosing techniques for buprenorphine, as reported by others, including the transition from the higher to lower dosing protocols [34].

It should be of note that, in our centers, multidimensional approaches have been used (other than pharmacology). Problematic substance use significantly contributes to morbidity and mortality and affects individuals and communities systematically. Responses to such issues requires a multidimensional, evidence-informed approach. Medical, psychological, and social responses have been identified as those with the most efficacies within the literature [35]; despite these considerations, discouraging the use of maintenance therapy in this disorder is still frequently reported in some states of U.S.A. [36].

## 5. Conclusions

Levomethadone appears to be promising in conditions with high cardiovascular risk, for instance, in the double diagnosis patients that are under treatment with antidepressant and antipsychotic drugs, which are known to cause cardiovascular adverse reactions with QTc prolongation. The effectiveness of levomethadone in maintenance therapy was observed at lower doses in males vs. females. Gender and race differences in the maintenance treatment, in favor of male vs. Latina females, have recently been reported [37], with an impact in the policy treatment in another reality [38].

All of these collected data will allow the construction of therapeutic algorithms that are useful for improving the prescription of these drugs in clinical practice, particularly recommended in complex patients, abusers, and/or polytherapy, as well as to increase the number of detox patients.

In this therapeutic area, some additional mu-opioid antagonists are available, such as naldemedine tosilate, methylnaltrexone, or the naloxegol PEG of naloxone, which was developed to overcome the side effects of naloxone. These drugs showed increased polarization, with respect to naltrexone or naloxone, with reduced permeability of the drug in the central nervous system (CNS), without interfering with central analgesic effects or withdrawal syndrome, but maintaining the effects as a high-affinity antagonist of opioid receptors in the gastrointestinal tract. In network meta-analysis, naloxone and naldemedine appear to be the most efficacious treatments for constipation, and naloxone was the safest of these agents [39]. Novel, long-acting buprenorphine formulations are also available. These long-acting buprenorphine formulations, such as the subdermal implant, as well as subcutaneous depot formulations, have the potential to reduce the risk of diversion [40]. The new depot buprenorphine formulations were found to be non-inferior to sublingual drugs, as well as more effective than a placebo, in reducing opioid use, and it is the preferred formulation vs. the sublingual one in the latest trial [41,42]. Depot buprenorphine, in custodial settings, showed treatment retention and outcomes, comparable to those observed in community settings and for other opioid agonist treatment used in custodial settings, without increased risk of diversion [43]. The cost-effectiveness of the innovative evaluation of buprenorphine implants was compared to the sublingual buprenorphine tablets over a 12-month simulated model and found the implant to be preferable, from a health-economic perspective, for the treatment of opioid dependence, as it was associated with lower total costs, more quality-adjusted life years (+0.031), and favorable incremental net monetary benefit, compared to the sublingual form [40]. The longer-acting formulations might also prove cost-effective, despite this need to be investigated in specific patient populations and settings.

Confounding factors may limit the significance of our conclusion; the low number of patients per subgroup requires further investigation. The risk of diversion, which is, however, expected to be low in our setting because the recruited patients were those adherent to therapy and collaborating with clinicians, should be also considered. The quality of life was not quantified in our study. This is reported to be superior with other drugs, rather than rac-methadone/levomethadone, as buprenorphine and diamorphine patients rated their physical functioning substantially higher than methadone patients [15]. Buprenorfine also showed more favorable metabolic effects vs. other drugs in these patients [44].

## Figures and Tables

**Figure 1 pharmaceutics-14-00461-f001:**
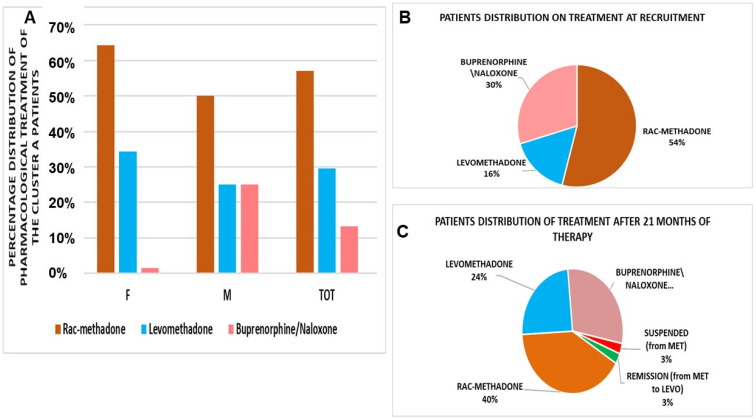
(**A**) Pharmacological treatment of the Cluster A patients (number of patients = 211). Rac-methadone, at the first visit, represented the preferred option. F = female, M = male, TOT = total aggregated data. Patient distribution on drug therapy at recruitment (**B**) and after 21 months of COVID-19, during the pandemic period. (**C**) Rac-methadone (MET), LEVO = levomethadone.

**Figure 2 pharmaceutics-14-00461-f002:**
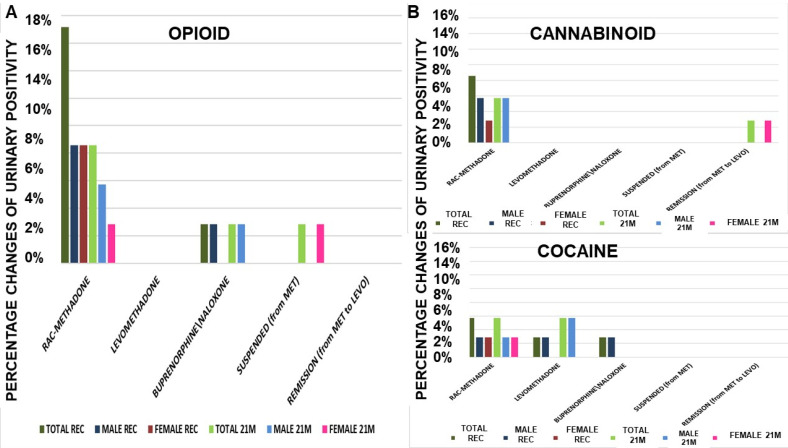
Opioids urine positivity at recruitment (REC) and after 21 months (21M) of therapy (**A**). Levomethadone was fully effective in preventing the cannabinoids positivity, but not rac-methadone and buprenorphine/naloxone. (**B**) Cannabinoids urine positivity at recruitment (REC) and after 21 months (21M) of therapy. Levomethadone and buprenorphine/naloxone were fully effective in preventing the cannabinoids positivity, but not rac-methadone. Cocaine urine positivity at recruitment (REC) and after 21 months (21M) of therapy. The treatments were less effective in preventing cocaine positivity. MET = rac-methadone, LEVO = levomethadone.

**Figure 3 pharmaceutics-14-00461-f003:**
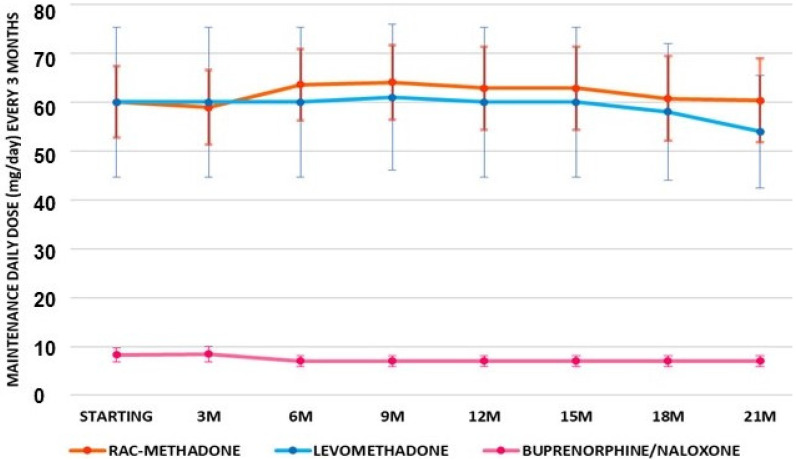
Maintenance means doses of patients under treatments (number of patients = 30) with rac-methadone, levomethadone, and buprenorphine-naloxone, at 90 days of intervals for 21 months (21M) of treatment.

**Figure 4 pharmaceutics-14-00461-f004:**
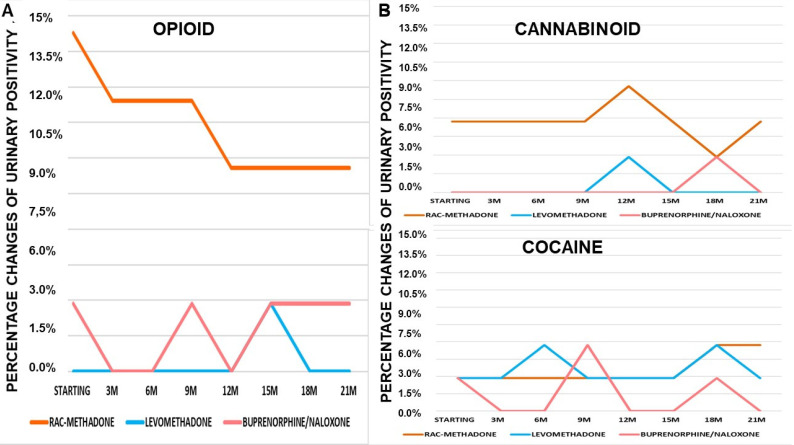
(**A**) Changes of opioids urinary positivity over 21 months (21M) of treatments. (**B**) Changes of cannabinoids urinary positivity over 21 months (21M) of treatments. Changes of cocaine urinary positivity over 21 months (21M) of treatments.

**Figure 5 pharmaceutics-14-00461-f005:**
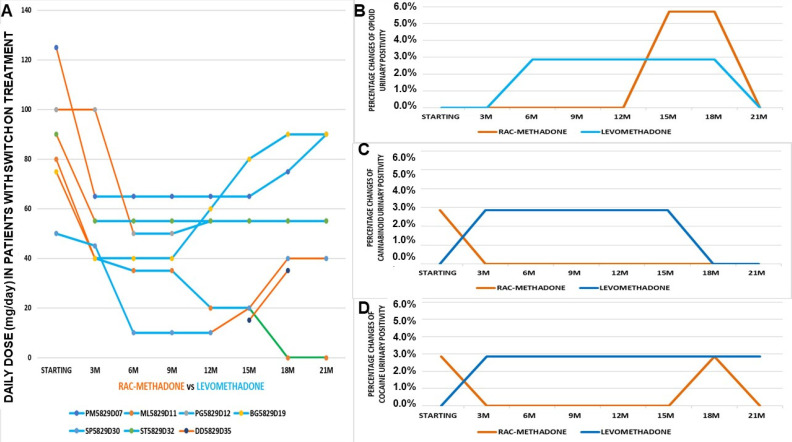
(**A**) Daily doses of patients with a switch of treatment over 21 months of monitoring (number of patients = 7). (**B**) Opioid positivity of patients, following a switch of treatment over 21 months of monitoring (number of patients = 2). (**C**) Cannabinoid positivity of patients, following a switch of treatment over 21 months of monitoring (number of patients = 2). (**D**) Cocaine positivity of patients with a switch of treatment over 21 months of monitoring (number of patients = 2).

**Figure 6 pharmaceutics-14-00461-f006:**
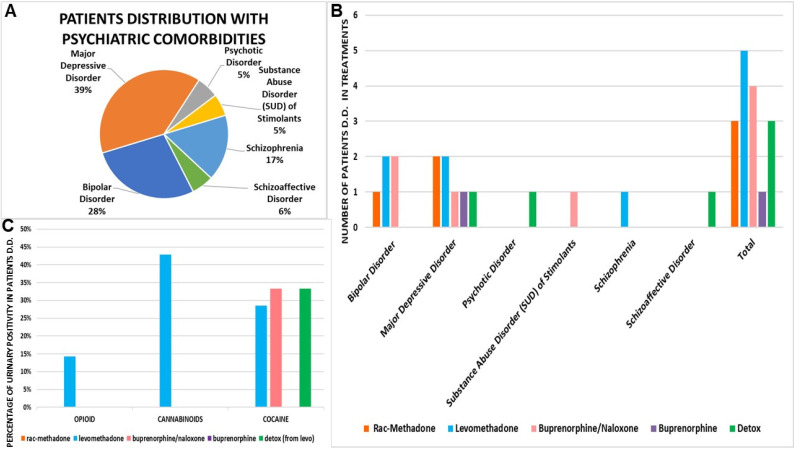
(**A**) Patients distribution with psychiatric comorbidities. (**B**) Drug treatment of patients with double diagnosis (D.D.). (**C**) Urinary positivity of patients with double diagnosis (D.D.) in treatments.

**Figure 7 pharmaceutics-14-00461-f007:**
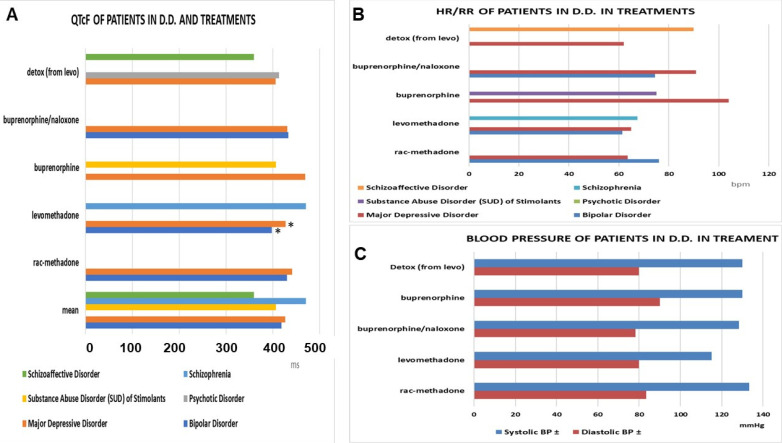
(**A**) The QTcF in the double diagnosis patients under maintenance treatments. Levomethadone showed the lowest values in bipolar patients and major depressive disorders (*) vs. other treatments (data significantly different * ANOVA one-way F > 2.5 *p* < 0.05), and the highest values in the schizophrenia patients, due to the inclusion of one congenital QT case. (**B**,**C**) Cardiovascular parameter in the double diagnosis (D.D.) patients under maintenance treatments. HR = heart rate, RR = intervals of the ECG.

**Table 1 pharmaceutics-14-00461-t001:** Phenotypic and demographic characteristics of the patients and gender distribution.

Patients	Number of Patients	Male	Female	Age	Body Max Index (BMI)
**Nationality:** Italian **Ethnicity:** Caucasian **Status:** Living alone (32%), living with parents or partner (68%), employed (30%), un-employed (35%), temporary job (35%)	266	212	54	44.80 ± 5.65	26.21 ± 3.83
**Cluster A**	211	171	40	44.71 ± 4.79	26.41 ± 3.00
**Cluster B**	37	27	10	43.54 ± 8.73	25.18 ± 3.16
**Cluster C**	18	14	4	48.44 ± 9.41	26.03 ± 4.94

**Table 2 pharmaceutics-14-00461-t002:** Phenotypic characteristics of the Cluster A patients and comorbidity distribution. VAS = visual analogue scale, as a score of craving level as a function of group and time; BMI = body mass index.

Patients	N Patients	Percentage of Patients	Age	BMI
**Psychiatric:** 20 ± 1.3% **Infectious:** 12 ± 5.9% **Cardiovascular:** 7 ± 5% **Metabolic:** 5 ± 0.9%	211	100%	44.17 ± 4.79	26.41 ± 3.00
**Male** **VAS** = 32/100	171	81%	44.41 ± 5.24	27.09 ± 4
**Female** **VAS** = 31/100	40	19%	43.16 ± 2.89	23.51 ± 5

**Table 3 pharmaceutics-14-00461-t003:** Drug treatments and dosages at patient recruitment.

Gender	Treatments	Daily Dose (mg/day)
**Female (19%)**	Rac-methadone	55 ± 10 mg/day
	Levomethadone	45.11 ± 6.5 mg/day
	Buprenorphine/naloxone	4 ± 2.62 mg/day
**Male (81%)**	Rac-methadone	57.21 ± 10.88 mg/day
	Levomethadone	28.75 ± 7.5 mg/day *°
	Buprenorphine/naloxone	8 ± 3.62 mg/day

* Data significantly different vs. rac-methadone group and ° female group.

**Table 4 pharmaceutics-14-00461-t004:** Drug effects on the urinary positivity of patients under treatment, after 180 days of treatment.

	Levomethadone N Patients = 62	Rac-Methadone N patients = 121	Buprenorphine/Naloxone N Patients = 28
**Heroin**	−53 ± 9% *	−41 ± 8% *	−42 ± 3.5% *
**Cannabinoids**	−48 ± 8% *	−32 ± 5% *	−49 ± 6% *
**Cocaine**	−37 ± 6% *	−35 ± 8% *	−36 ± 9% *

* Data significantly different vs. data at recruitment by Student’s *t*-test for *p* < 0.05. No differences within and between groups were observed using ANOVA (F = 1.09) for *p* < 0.05.

**Table 5 pharmaceutics-14-00461-t005:** QT intervals of the ECG trace, corrected by Fridericia formula (QTcF) at recruitment and after 180 days of treatment on 123 patients. No differences, within and between groups, were observed using ANOVA, with F values close to 1 for all parameters, for *p* < 0.05, at recruitment and after 180 days of treatment.

	Recruitment	Levomethadone N Patients = 35	Rac-Methadone N Patients = 60	Buprenorphine/ Naloxone N Patients = 28
**QTcF (ms)**	430.2 ± 12.4	425.1 ± 7.4	444.5 ± 9.2	428.1 ± 9.4
**Heart rate (bpm)**	75.23 ± 5.3	76.13 ± 7.3	77.13 ± 6.3	76.13 ± 7.3
**Systolic blood pressure (mmHg)**	122.5 ± 14	129.5 ± 13	127.5 ± 11	124.5 ± 12
**Diastolic blood pressure (mmHg)**	81.4 ± 3.3	79.4 ± 8.3	81.1 ± 8	79.1 ± 9
	**After 180 Days**	**Levomethadone** **N Patients = 35**	**Rac-Methadone** **N Patients = 60**	**Buprenorphine/** **Naloxone** **N Patients = 28**
**QTcF (ms)**	431.77 ± 8.83	426 ± 8.4	439.1 ± 7.1	430.2 ± 23
**Heart rate (bpm)**	78.17 ± 6.53	79.1 ± 9	78.1 ± 4.3	77.3 ± 6.3
**Systolic blood pressure (mmHg)**	124.9 ± 11	123.1 ± 15	128.1 ± 13	123.5 ± 9
**Diastolic blood pressure (mmHg)**	78.07 ± 10	77.1 ± 10	78.1 ± 9	79 ± 11

**Table 6 pharmaceutics-14-00461-t006:** Socio-demographic characteristics of the patients.

	Number of Patients	Percentage	Age±	BMI±
**Total number of patients**	37	100%	43.54 ± 4	25.18 ± 3.16
**Male**	27	72.97%	44.93 ± 6	25.25 ± 3.06
**Female**	10	27.03%	39.80 ± 8	24.98 ± 3.27
**Residence conditions:**				
Family of origin (with parents or high relatives)			15	
Newly formed family (with spouse, partner, children, etc.)			17	
Alone			5	
**Previous:**				
Criminal record (reports, arrest, house arrest, prison, etc.)			10	
Therapeutic communities			10	
Both			8	
None			9	

**Table 7 pharmaceutics-14-00461-t007:** Drug treatment after 21 months of observations, during COVID-19.

Drugs at Recruitment	Medium Dose (mg/day)	Male Dose (mg/day)	Female Dose (mg/day)
**Rac-methadone**	66.25 ± 30.82	71.15 ± 31.27	57.14 ± 27.76
**Levomethadone**	58.33 ± 31.58	58.33 ± 31.58	No patients
**Buprenorphine\naloxone**	8.27 ± 4.94	8.88 ± 5.18	6.67 ± 3.77
**Drugs after 21 Months**	**Male Dose (mg/day)**	**Female Dose (mg/day)**	**Medium Dose (mg/day)**
**Rac-methadone**	59.00 ± 31.42	55.50 ± 34.89	66.00 ± 21.31
**Levomethadone**	62.22 ± 24.28	62.22 ± 24.28	No patients
**Buprenorphine\naloxone**	7.00 ± 3.79	7.13 ± 3.79	6.61 ± 2.17

**Table 8 pharmaceutics-14-00461-t008:** Characteristics of the patients. BMI = body mass index.

Patients (N = 18) Age= 48.4 ± 10; BMI =26.0 ± 6.41 Dose	Rac-Methadone Dose (mg/day) (Number of Patients = 3) 97.5 ± 8	Levomethadone Dose (mg/day) (Number of Patients = 10) 51.8 ± 5	Buprenorphine/Naloxone Dose (mg/day) (Number of Patients = 3) 2.5 ± 1	Buprenorphine Dose (mg/day) (Number of Patients = 2) 3
Female (N = 4) Age = 49.75 ± 12 BMI = 25.09 ± 9	60 ± 4	65 ± 3	1 ± 0.4	/
Male (N = 14) Age = 48.07 ± 8.4; BMI = 26.96 ± 3.7	135 ± 11	38.6 ± 4	4 ± 1	3

**Table 9 pharmaceutics-14-00461-t009:** Psychological indicators of disease. SCL90 = Symptom Checklist-90, a questionnaire used to assess psychological problems of 90 items.

Drugs	SCL90 Global Severity Index (GSI) ±	Positive Symptom Total (PST)	SCL90 Positive Symptom Distress Index (PSDI)
**Rac-Methadone**	1.61 ± 0.3	64.5 ± 4	2.27 ± 0.1
**Levomethadone**	1.59 ± 0.7	61 ± 9	2.26 ± 1
**Buprenorphine/Naloxone**	0.67 ± 0.09	35.5 ± 8	1.7 ± 1
**Buprenorphine**	2.74 ± 0.9	78 ± 11	3.17 ± 0.8
**Levomethadone Detox**	0.81 ± 0.2	33 ± 8	2.21 ± 0.9

**Table 10 pharmaceutics-14-00461-t010:** The medical condition of recruited patients with a double diagnosis. QTcF = QT interval of the ECG trace (corrected by the Fridericia formula), BP = blood pressure, bpm = beat per minute, HR = heart rate, RR = RR intervals of the ECG, BMI = body mass index, GSI = global severity index, PST = positive symptom total, PSDI = positive symptom distress index.

Medical Conditions	Number of Patients	Maintenance Treatments	Concomitant Drugs
**Total patients** Female (N = 4) Age = 49.75 ± 12.84 BMI = 25.09 ± 9.05 Male (N = 14) Age= 48.07 ± 8.43 BMI = 26.96 ± 3.77	18		Antidepressant, antipsycotic
**Blood pressure (BP) (systolic/diastolic)**			
133.33 ± 21/83 ± 9 mmHg	3	Rac-methadone	
115.1 ± 18/80 ± 5	10	Levomethadone	
128.2 ± 18/78 ± 6	3	Buprenorphine/naloxone	
130.6 ± 15/90 ± 4	2	Buprenorphine	
130.1 ± 11/80 ± 6	3	Levomethadone detox	
HR/RR			
67.67 ± 10 bpm/ms	3	Rac-methadone	
64.67 ± 6	10	Levomethadone	
80 ± 7	3	Buprenorphine/naloxone	
89.5 ± 8	2	Buprenorphine	
76 ± 9	3	Levomethadone detox	
QTcF			
438.67 ± 7.57 ms	3	Rac-methadone	
437.71 ± 40.6	10	Levomethadone	
432.33 ± 2.31	3	Buprenorphine/naloxone	
438 ± 43.84	2	Buprenorphine	
383 ± 35.53	3	Levomethadone detox	
**bipolar disorder** **Substance use disorder (DUS)** BMI = 28.56 ± 4.92 BP= 120.00 ± 11/75 ± 8 mmHg QTcF = 418.40 ± 18.93 HR/RR = 69.60 ± 9 bpm/ms	5 (4 of type 2 and 1 of type 1)		
QTcF = 433 ± 12 ms, HR/RR = 74.5 ± 13 bpm/ms BP = 120 ± 11/75 ± 8 mmHg urine heroin, cocaine, amphetamine negative, one case of cannabinoid positivity; GSI 0.59, PST 30, PSDI 1.77, urine negative	2	Buprenorphine/naloxone 3.5 ± 1 mg/day	Sertalin, litium, quetiapine, flurazepam, bupropione, sodium valproate, alipiprazol
QTcF = 430 ms, HR/RR = 76 bpm/ms BP = 130/80 mmHg urine heroin, cocaine, amphetamine negative; GSI 1.63, PST 59, PSDI 2.49	1	Rac-methadone 60 mg/day	Vortioxetine, aripripazole, prazepam, zolpidem
QTcF = 394 ms, HR/RR = 58 bpm/ms, HCV+, cannabinoids and cocaine+, cannabis and bzd abused; urine cannabis, cocaine+; GSI 0.93, PST 45, PSDI 1.87	1	Levomethadone 80 mg/day	Aripiprazole, asenapine
QTcF = 402 ms, HR/RR = 65 bpm/ms, HCV+, urine heroin, cocaine, amphetamine negative; GSI/, PST/, PSDI/	1	Levomethadone switched from rac-methadone to levomethadone 15 mg/day and after one year to low dose 3 mg/day of levomethadone	Aripiprazole, valproate, clorpromazine
**Major depressive disorder** **Substance Use Disorder (DUS)** BMI = 23.32 ± 8 BP = 134 ± 21/84.86 ± 12 mmHg QTcF = 426.80 ± 17.51 ms HR/RR = 71.20 ± 19 bpm/ms	7		
QTcF = 427.5 ± 18 ms, HR/RR= 65 ± 9 bpm/ms, BP = 130 ± 11/94.96 ± 10 mmHg HCV + 1a urine heroin, cocaine, amphetamine, cannabinoids negatives, one case of codeine positivity GSI 1,92 ± 0.1, PST 69 ± 8, PSDI 2.46 ± 0.3	2	Levomethadone 90 ± 9 mg/die	Duloxetine, valproate, levosulpiride, clonazepam, pregabalin
QTcF = 406 ms, HR/RR = 62 bpm/ms, BP = 130/94.96 mmHg HCV + , urine heroin, cocaine, amphetamine, cannabinoids negatives	1	Levomethadone starting dose 35 mg/day and then 3 mg/day until completed detox	Valproate, fluoxetine, olanzapine, levomepromazine, estazolam
QTcF = 443 ± 2 ms, HR/RR = 63.5 ± 9 bpm/ms, BP = 131 ± 13/91.86 ± 10 mmHg (N patient =1 HIV+, HCV +) urine heroin, cocaine, amphetamine, cannabinoids negatives One case of severe hepatopathies One case of positivity to benzodiazepine One case of GSI 1.59, PST 70, PSDI 2.04	2	Rac-methadone 105 ± 12 mg/die	Valproate, sertalin, zolpidem, lithium, duloxetine, pregabalin, clonazepam ritonavir, atazanavir, emtricitabine, tenofovir, disoproxil
QTcF = 469 ms, HR/RR = 104 bpm/ms, BP = 129/90.81 mmHg, urine heroin, cocaine, amphetamine, cannabinoids negatives, GSI 2.74, PST 78, PSDI 3.17	1	Buprenorfine 2 mg/die	Litium, olanzepine, levomepromazin, clonazepam
QTcF = 431 ms, HR/RR = 91 bpm/ms, BP = 128/89 mmHg, urine heroin, cocaine, amphetamine, cannabinoids negatives GSI 0.74, PST 41, PSDI 1.63	1	Buprenorphine/naloxone 8 mg/die	Aripiprazole, valproate, topiramate, trazodone (ER)
**Psychoses** BMI 23.99 QTcF = 413 ms, HR/RR = 91 bpm/ms, BP = 110/70 mmHg urine heroin, amphetamine, cannabinoids negatives cocaine positive	1	Levomethadone 60 mg/day until completed detox	Duloxetina, trazodone, clonazepam, quetiapina, flurazepam
**Schizophrenia** BMI = 26.79 ± 6	3		
QTcF = 471.00 ± 36.51 ms, HR/RR = 67.5 ± 12 bpm/ms, BP = 116.5 ± 11/83.34 ± 8 mmHg One case of congenital LQT = 500 ms urine heroin, cocaine, amphetamine, cannabinoids negative, and two cases of cannabinoid positive and one case of cocaine positive GSI/, PST/, PSDI/	3	Levomethadone 50 ± 4 mg/day and one case of rac-methadone 120 mg/die switch to levomethadone 80 mg/die	Risperidone, lurasidone, promazine, clonazepam, aloperidol, flurazepam, aripirazolo, sertralin
**Schizo-affective disorder** BMI = 34.68 BP= 140/90 mmHg QTcF = 360 ms, HR/RR = 90 bpm/ms, urine heroin, cocaine, amphetamine, cannabinoids negative, GSI 0.81, PST 33, PSDI 2.21.	1	Rac-methadone 60 mg/die and switch to levomethadone starting dose 30 mg/day and then 3 mg/day until completed detox	Valproate, levomepromazine, diazepam, biberidene, quetiapine, flurazepam
**DUS cocaine** BMI = 32.66 BP = 130/10 mmHg QTcF = 407 ms, HR/RR = 75 bpm/ms urine/ GSI/, PST/, PSDI/	1	Buprenorphine 4 mg/die	Lithium carbonate, aripiprazole, clonazepam

## Data Availability

Data supporting the reported results are available for further evaluation.

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
