# Peer review of "Multicenter Observational/Exploratory Study Addressed to the Evaluation of the Effectiveness and Safety of Pharmacological Therapy in Opioid-Dependent Patients in Maintenance Therapy in Southern Italy"

_pharmaceutics, 2022, doi:10.3390/pharmaceutics14020461_

Round 1

Reviewer 1 Report

The discussion must be improved, the authors only present results without any reflections about their significance and limitations. What about some confounding factors?. The methodology looks adequate, the statistical analysis is correct, the introduction could be less wide, removing some definitions well known.

Author Response

We thanks the reviewer for her/his positive comments. The Introduction was shorted and the discussion was extended also considering the limitation of our work and confounding factors in the revised vs of the manuscript.

Reviewer 2 Report

The topic of this article is interesting, the authors presenting the results of a multicenter-observational study performed in the Southern Italy to assess the effectiveness of rac-methadone, levomethadone and buprenorphine in opioid-dependent patients with polymedication for comorbidities.

After reading the manuscript, the following doubts and suggestions have arisen.

The introduction and the discussion sections should be more complete, providing supplementary background about the recent communicated data on the use of rac-methadone, levomethadone and buprenorphine in opioid addicted patients.

The results obtained should be compared with those achieved by other researchers and discussions should be significantly detailed.

Literature analysis reveals others communicated data in the field should be cited, for example:

  • Askari MS et al. Medication for opioid use disorder treatment and specialty outpatient substance use treatment outcomes: differences in retention and completion among opioid-related discharges in 2016. J Subst Abus Treat. 2020;114:108028.
  • Beetham T et al. Therapies offered at residential addiction treatment programs in the united states. JAMA. 2020;324(8):804–6.
  • Dunlop AJ et al. Treatment of opioid dependence with depot buprenorphine (CAM2038) in custodial settings. Addiction. 2021 Jun 29. doi: 10.1111/add.15627.
  • Dydyk AM et al. Opioid Use Disorder. [Updated 2021 Jul 12]. In: StatPearls [Internet]. Treasure Island (FL): StatPearls Publishing; 2021 Jan-. Available from: https://www.ncbi.nlm.nih.gov/books/NBK553166/
  • Elman I et al. Metabolic and Addiction Indices in Patients on Opioid Agonist Medication-Assisted Treatment: A Comparison of Buprenorphine and Methadone. Sci Rep10, 5617 (2020).
  • Frank D et al. “It’s like ‘liquid handcuffs”: The effects of take-home dosing policies on Methadone Maintenance Treatment (MMT) patients’ lives. Harm Reduct J18, 88 (2021).
  • Guerrero E et al. Gender disparities in opioid treatment progress in methadone versus counseling. Subst Abuse Treat Prev Policy16, 52 (2021).
  • Kumar R et al. Buprenorphine. [Updated 2021 Aug 6]. In: StatPearls [Internet]. Treasure Island (FL): StatPearls Publishing; 2021 Jan-. Available from: https://www.ncbi.nlm.nih.gov/books/NBK459126/
  • Lintzeris N et al. Patient-Reported Outcomes of Treatment of Opioid Dependence With Weekly and Monthly Subcutaneous Depot vs Daily Sublingual Buprenorphine: A Randomized Clinical Trial. JAMA Netw Open.2021;4(5):e219041.
  • Mahon D. Improving retention in opioid treatment. J Addict Addictv Disord. 2020;7:47.
  • Figgatt MC et al. Take-home dosing experiences among persons receiving methadone maintenance treatment during COVID-19, Journal of Substance Abuse Treatment, 2021; 123, 108276.
  • Deutsch-Link S et al. (2021)Race-based differences in drug use prior to onset of opioid use disorder, Journal of Ethnicity in Substance Abuse, DOI: 1080/15332640.2021.1879702
  • Soyka M. Transition From Full Mu Opioid Agonists to Buprenorphine in Opioid Dependent Patients—A Critical Review. Frontiers in Pharmacology 2021; 12, 2901.
  • Guillery SPE et al. Quality of Life in Opioid Replacement Therapy: A Naturalistic Cross-Sectional Comparison of Methadone/Levomethadone, Buprenorphine, and Diamorphine Patients. Eur Addict Res. 2021;27(5):371-380. doi: 10.1159/000514192.

Some other aspects were found in this manuscript:

- different fonts were used in the text and in the figures;

- a lot of abbreviations were not explained (i.e. QtcF, VAS, NMDA, C.R.F., MEDRA, ASI, Scl90, ADRs, ECG, TOT, 1Y9M, HR/RR, OUD, OPI+, CAN+, COC+, CNS, BUP), which causes a large part of the transmitted information to be difficult to understand;

- at line 584 ``SCORE`` should be written in lowercase;

- spelling check of the text is mandatory.

- the authors should upgrade the references;

- English including grammar, style and syntax, should be extensively improved through the professional help from English Editing Company for Scientific Writings.

Author Response

  • We thank the reviewer for her/his positive comments. The discussion extended taking into account the recent results and literature data suggested by the reviewer comparing the Regional data with the international data. The reference list was therefore updated with all references suggested that were appropriately proposed by the reviewer.

Round 2

Reviewer 2 Report

I received the manuscript in which the authors made the changes according to the reviewers recommendations and suggestions.

It consider it could be accepted for publication in this journal, but I propose to have the manuscript checked by a native English speaking person.